# The Effects of Chronic Unpredictable Mild Stress and Semi-Pure Diets on the Brain, Gut and Adrenal Medulla in C57BL6 Mice

**DOI:** 10.3390/ijms241914618

**Published:** 2023-09-27

**Authors:** Mauritz Frederick Herselman, Larisa Bobrovskaya

**Affiliations:** Health and Biomedical Innovation, Clinical and Health Sciences, University of South Australia, Adelaide, SA 5000, Australia; mauritz.herselman@mymail.unisa.edu.au

**Keywords:** stress, depression, diet, gut-brain axis, microbiota, pathways, diet

## Abstract

Chronic stress is known to perturb serotonergic regulation in the brain, leading to mood, learning and memory impairments and increasing the risk of developing mood disorders. The influence of the gut microbiota on serotonergic regulation in the brain has received increased attention recently, justifying the investigation of the role of diet on the gut and the brain in mood disorders. Here, using a 4-week chronic unpredictable mild stress (CUMS) model in mice, we aimed to investigate the effects of a high-fat high-glycaemic index (HFD) and high-fibre fruit & vegetable “superfood” (SUP) modifications of a semi-pure AIN93M diet on behaviour, serotonin synthesis and metabolism pathway regulation in the brain and the gut, as well as the gut microbiota and the peripheral adrenal medullary system. CUMS induced anxiety-like behaviour, dysregulated the tryptophan and serotonin metabolic pathways in the hippocampus, prefrontal cortex, and colon, and altered the composition of the gut microbiota. CUMS reduced the catecholamine synthetic capacity of the adrenal glands. Differential effects were found in these parameters in the HFD and SUP diet. Thus, dietary modifications may profoundly affect the multiple dynamic systems involved in mood disorders.

## 1. Introduction

Although stress is a part of everyday life and many organisms have evolved homeostatic mechanisms to manage acute stress events, chronic stress exposure is known to lead to maladaptive responses of the major stress systems: the hypothalamic-pituitary-adrenal (HPA) axis responsible for the release of cortisol, and the sympathoadrenal medullary system for the release of adrenaline [1,2,3]. This maladaptation can lead to perturbations in the mesolimbic systems of the brain, resulting in impairments in learning and memory, mood and decreased reward sensitivity, increasing the risk of developing mood disorders such as generalised anxiety disorder and depression [2,3].

The study of molecular mechanisms underlying the involvement of the mesolimbic system in anxiety, and even more so in depression, has had a focus on the monoamine neurotransmitter concentrations in brain regions such as the hippocampus and prefrontal cortex (PFC) for decades, and this led to the development of antidepressant medications such as serotonin reuptake inhibitors which act on the serotonin transporter (SERT) to inhibit the re-uptake of serotonin [4,5]. Several other theories of depression have since been proposed involving other major neurotransmitters, such as dopamine and noradrenaline [6], as well as the recognition of the influence of the hypothalamic-pituitary-adrenal axis [7]. Many pre-clinical animal studies show that monoamine neurotransmitter regulation is altered in stress and depression-like states [8,9,10,11], however the chronic unpredictable mild stress (CUMS) model is most widely used to robustly induce behavioural endpoints of anxiety and depression-like behaviours in mice [12]. Recently, the bidirectional gut-brain axis has gained increased attention in neuropsychiatric disorders due to the close association of intestinal symptoms with these disorders. It is also known that monoamine neurotransmitters in the brain are produced from essential amino acids derived from the diet and resident microorganisms of the gut [13,14], further strengthening this association. Both pre-clinical and clinical studies show evidence of alterations in the gut microbiota in chronic stress and depression, and the gut microbiota is known to modulate dietary tryptophan metabolism, the key amino acid in the serotonin synthesis pathway [5,15,16,17,18,19].

The serotonin metabolic pathway (Figure 1) plays a crucial role in tryptophan metabolism by synthesising serotonin via tryptophan hydroxylase 2 (TPH2) [5]. Thereafter, along with all other monoamine neurotransmitters, serotonin is packaged into vesicles by vesicular monoamine transporter 2 (VMAT2), ready for synaptic release [20]. Monoamines released into the synaptic cleft are regulated by reuptake via transporters such as SERT and can be further regulated pre-synaptically by degradation through monoamine oxidase-A (MAO-A) [5,14]. Tryptophan metabolism can be “shunted” away from the serotonin pathway and down the kynurenine pathway by indoleamine 2,3-dioxygenase (IDO), leading to the production of neuroprotective kynurenic acid and neurotoxic quinolinic acid [4,14]. The enzymes involved in the serotonin and kynurenine pathways are found in both the brain and the gut, although peripheral tissues more dominantly express the isoform tryptophan hydroxylase 1 (TPH1) than TPH2 [5,14]. Importantly, both the serotonin and kynurenine pathways have been implicated in depression, evidenced by increased IDO levels in the brain and the gut with increased neurotoxic quinolinic acid and kynurenic acid in the brain and the gut, respectively [8], as well as increased MAO-A levels with decreased TPH1/2 levels in the brain and gut [21,22].

In addition to monoamines in the brain, reduced mature brain-derived neurotrophic factor (mBDNF) levels have also been implicated in chronic stress and depression as a cause of reduced neuroplasticity [23]. mBDNF levels have also been positively correlated with dietary tryptophan content [14]. Furthermore, dysregulation of the cross-talk between serotonin and BDNF has been implicated in the development of the disorder [24], with dysregulation of the HPA axis possibly driving this process [23]. Interestingly, the gut microbiota is also known to modulate the HPA axis, widely studied in the CUMS model [12,25]. However, studies on the effects of CUMS on the sympathoadrenal medullary system are sparse, even though there is evidence of high expression levels of SERT in the adrenal medulla [26], potentially playing a role in modulating catecholamine secretion during stress response. The synthesis of adrenaline and noradrenaline in the adrenal medulla is rate-limited by tyrosine hydroxylase (TH), and studies show that acute stress activates TH through phosphorylation [27,28]. However, TH activation in chronic stress is not well studied. Furthering our understanding of the mechanisms of the sympathoadrenal medullary system is important since there is evidence that this system is blunted in depressive states [29]. Therefore, the study of molecular mechanisms contributing to psychiatric disorders such as anxiety and depression requires appreciation and consideration of the multiple dynamic systems involved.

Importantly, spontaneous remission of disorders brought about by chronic stress is unlikely without intervention [2]. While many current monoamine-based antidepressants are effective, most have also shown frequent relapse, low efficacy and delayed effects, which may be attributed to the complexity of chronic stress disorders [30]. Recently, dietary interventions in mood disorders have received increasing attention due to the interactions between the brain, the gut and the gut microbiota [31]. There is evidence that “Westernised” diets, high in carbohydrates and saturated fats, are associated with depression as they can impair brain systems and negatively affect gut health [32,33]. Plant-based, often called “superfoods”, such as cacao and blueberries, have positively affected both brain systems and gut health and reduced inflammation [34,35,36]. The majority of studies have used preclinical animal models of chronic stress with non-purified “standard chow” diets to investigate these bidirectional gut-brain mechanisms, possibly contributing to the variability in reported findings since non-purified diets are not standardised across the field of neuroscience [37,38]. According to Lee et al. [37], “standard chow” diets can vary considerably in energy and vitamin content from different manufacturers. Thus, it is of utmost importance that rodent studies investigating dietary interventions are carried out with purified diets to ensure data reproducibility. We hypothesised that mice fed a semi-pure AIN93M standard diet would produce anxiety-like and depressive-like behaviours following CUMS due to the widespread use of this protocol as a chronic stress model and that these behavioural endpoints could be manipulated through either plant-based or “Westernised” modifications of the AIN93M diet. Therefore, the present study sought to evaluate the effects of a high-fat high-glycaemic index “Westernised” (HFD) and a high fibre fruit & vegetable “superfood” (SUP) modification of a semi-pure AIN93M rodent diet on behaviour, the regulation of tryptophan metabolism in the brain and colon, as well as on gut health and the gut microbiota in a CUMS model in C57BL6 mice. This study also aimed to investigate the effects of CUMS, and these diets on the regulation of adrenomedullary TH.

## 2. Results

### 2.1. Body Weight

The body weight of each mouse was measured weekly during the study, as shown in Figure 1. Two-way ANOVA on the final weight of the mice in each group showed a significant main effect of diet (*p* < 0.0001, main effect; *p* < 0.0001, AIN93M vs. HFD; *p* = 0.0202, AIN93M vs. SUP; *p* < 0.0001, HFD vs. SUP), with increased body weight in the control HFD compared with the control AIN93M group (4.560 ± 0.97, *p* < 0.0001) and compared with the control SUP group (6.380 ± 0.97, *p* < 0.0001). There were no effects of CUMS treatment on final body weight and, similarly to the control, a significant increase in body weight was found in the CUMS HFD group compared with the CUMS AIN93M group (5.042 ± 0.92, *p* < 0.0001) and compared with the CUMS SUP group (6.960 ± 0.90, *p* < 0.0001). These results show that the administration of the HFD resulted in rapid weight gain by the end of the study protocol, while the body weight in the SUP diet and AIN93M diet did not differ significantly from each other, although the mice on the SUP diet had slightly lower body weight. Overall, CUMS did not have a significant effect on body weight in any of the diets.

### 2.2. General Biochemistry

Serum levels of triglycerides, cholesterol, total protein, and albumin were measured to assess dietary-related general blood biochemistry. The mean levels of cholesterol, total protein and albumin are presented in Figure 2. The data for serum triglycerides (see Appendix A) did not follow Gaussian distribution. Thus, the data was inversely transformed before analysis. Analysis of transformed serum triglyceride levels revealed a significant main effect of diet (*p* = 0.0003, main effect; *p* = 0.0018, AIN93M vs. SUP; *p* = 0.0006, HFD vs. SUP; inverse transformation of data shown). No significant differences were found between any of AIN93M groups compared with the HFD groups; however, a decrease was evident for the SUP control group compared to both the AIN93M control (−1.322 ± 0.44, *p* = 0.0172; inverse transformation of data shown) and HFD control groups (−1.548 ± 0.47, *p* = 0.0071; inverse transformation of data shown), suggesting that the SUP diet lowered serum triglyceride levels.

Analysis of serum cholesterol levels revealed a significant main effect of diet (*p* < 0.0001, main effect; *p* = 0.0045, AIN93M vs. HFD; *p* = 0.0235, AIN93M vs. SUP; *p* < 0.0001, HFD vs. SUP), with a significant increase found in the control HFD group compared with the control AIN93M group (1.416 ± 0.51; *p* = 0.0263). A significant reduction in cholesterol levels was found in the control SUP group compared with the control HFD group (−2.448 ± 0.51; *p* < 0.0001) as well as between the CUMS SUP and CUMS HFD groups (−2.084 ± 0.51; *p* = 0.0008). These results suggest that the HFD increased cholesterol levels, while the SUP diet reduced cholesterol. Analysis of total protein levels in serum showed no significant differences across any of the groups. While the same was found for serum albumin, a near significant interaction effect was found (*p* = 0.0711), with trends suggesting that CUMS slightly reduced albumin levels on the AIN93M diet, while CUMS slightly increased levels on the SUP diet, although neither reached significance.

### 2.3. Behaviour

#### 2.3.1. Open Field Test

Anxiety-like behaviour and locomotion were assessed using the open field test (OFT) (Figure 3). Significant main effects of CUMS (*p* = 0.0037) and diet (*p* = 0.0400, main effect; *p* = 0.0400, AIN93M vs. SUP) was found for the measure of total distance in the OFT. No significant differences were found between any groups except for a decrease in the CUMS SUP group compared with the control SUP group (−3.225 ± 1.86, *p* = 0.052), suggesting that this group of mice experienced a reduction in locomotive function.

Since the data for the time spent in the central zone did not follow Gaussian distribution, it was logarithmically transformed before analysis. Analysis of the transformed time spent in the central zone see Appendix A showed a significant main effect of CUMS (*p* = 0.0005) but no main effects of diet, suggesting that CUMS induced anxiety-like behaviour regardless of diet. It was further shown that a significant decrease in time spent occurred in the CUMS SUP group compared with the control SUP group (−0.3065 ± 0.14, *p* = 0.0442)

Since locomotive differences existed between the groups, the results of entries into the central zone were corrected per Total Distance before analysis. A significant main effect of CUMS (*p* = 0.0004) was found, with central zone entries reduced in both the AIN93M and SUP CUMS groups compared with control counterparts (−0.4127 ± 0.13, *p* = 0.0100, AIN93M; −0.2981 ± 0.13, *p* = 0.0818, SUP). Similar results were found for entries into the peripheral zone with a main effect of CUMS (*p* = 0.0018) and a reduction in the number of entries in the CUMS AIN93M compared to the control AIN93M group (−0.3630 ± 0.13, *p* = 0.0226). Altogether, these results suggest that CUMS reduced exploratory behaviour in the open field. The SUP diet also appeared to have this effect, while the HFD did not appear to have any effect on measures of the OFT.

#### 2.3.2. Elevated Plus-Maze Test

Anxiety-like behaviours were further investigated using the elevate plus-maze test (EPM) (Figure 4). Measures of total distance showed a significant main effect of diet (*p* < 0.0001, main effect; *p* < 0.0001, AIN93M vs. SUP; *p* = 0.0044 HFD vs. SUP), with a reduction in the distance travelled in the control SUP group versus the control AIN93M group (−3.537 ± 1.10, *p* = 0.0060), and in the control SUP versus the control HFD (−3.072 ± 1.10, *p* = 0.0202). A reduction in the distance travelled was also found in the CUMS SUP group versus CUMS AIN93M group (3.419 ± 1.07, *p* = 0.0067). These results suggest that the SUP diet reduced locomotion and/or exploratory behaviour in the EPM.

Since locomotive differences existed between the groups, the results of entries into the open arms and entries into the closed arms were corrected per Total Distance before analysis. No significant differences between the diets or groups were found for the open-arm entries. For the closed-arm entries, a significant main effect of CUMS was found (*p* = 0.0142), with no significant differences between the control and CUMS groups except in the SUP diet (0.2201 ± 0.09, *p* = 0.0478). The time spent in each arm was also evaluated. For the time spent in the open arms, a significant main effect of diet was found (*p* = 0.0019, main effect; *p* = 0.0013, AIN93M vs. SUP), and a reduction in the time spent in the open arms was found in control SUP group compared with the control AIN93M group (−14.72 ± 5.48, *p* = 0.0284) and between the CUMS SUP and the CUMS AIN93M groups (−13.89 ± 5.36, *p* = 0.0360). Similarly, analysis of time spent in the closed arms revealed a significant main effect of diet (*p* = 0.0006, main effect; *p* = 0.0006, AIN93M vs. SUP; *p* = 0.0228, HFD vs. SUP). An increase in the time spent in the closed arms was found in the control SUP group versus the control AIN93M group (39.71 ± 13.10, *p* = 0.0109) and between the CUMS SUP and CUMS AIN93M groups (33.55 ± 12.79, *p* = 0.0336). Time in the central zone also showed a main effect of diet (*p* = 0.0024, main effect; *p* = 0.022, AIN93M vs. SUP; *p* = 0.0507, HFD vs. SUP). The amount of time spent in the central zone was significantly reduced in the control SUP group compared with the control AIN93M (−18.63 ± 7.16, *p* = 0.0352) and in the CUMS SUP group compared with the CUMS AIN93M (−14.14 ± 7.00, *p* = 0.0519).

Taken together, these results suggest that the CUMS did not markedly increase anxiety-like behaviour in the EPM. However, the increased entries into the closed arms suggest that the mice in the CUMS group may have been slightly more anxious than controls. The SUP diet appeared to increase anxiety-like behaviour, since the greatest increase in closed-arm entries occurred in the CUMS SUP group. However, the SUP likely decreased exploratory behaviour overall due to the decreased total distance travelled by mice on this diet. This decrease in exploratory behaviour in SUP mice may have also been due to the high fibre levels of the diet causing discomfort, thus reducing locomotion in these mice.

#### 2.3.3. Tail Suspension Test

The tail suspension test (TST) assessed depressive-like behaviour (Figure 5). Analysis of the immobility time revealed a significant main effect of diet (*p* = 0.0235, main effect; *p* = 0.0802, AIN93M vs. SUP; *p* = 0.0366, HFD vs. SUP), with a reduction found in the CUMS SUP group compared with the CUMS AIN93M group (−46.31 ± 17.02, *p* = 0.0269), suggesting that the SUP diet may have attenuated behavioural despair or depressive-like behaviours.

### 2.4. Serotonin Synthesis and Metabolism Pathways in the Hippocampus and Prefrontal Cortex

The regulation of serotonin synthesis and metabolism pathways in response to CUMS and/or diet in the hippocampus (Figure 6) and prefrontal cortex (Figure 7) are shown. In the hippocampus, two-way ANOVA of TPH2 levels revealed a significant main effect of CUMS (*p* = 0.0090) and a significant interaction effect between the diets and treatment (*p* = 0.0074). TPH2 levels were increased in the SUP control group compared with the AIN93M control group (0.2577 ± 0.10, *p* = 0.0438) and increased in the AIN93M CUMS group compared to the respective control (0.4386 ± 0.10, *p* = 0.0004). 

Analysis of IDO levels showed a significant main effect of CUMS (*p* = 0.0022), with an increase in IDO levels in the CUMS AIN93M group compared to the CUMS control group (0.4199 ± 0.15, *p* = 0.0208); While a significant main effect of CUMS was found for MAO-A levels in the hippocampus (*p* = 0.0422). The results for IDO show that CUMS increased levels, especially in mice on the AIN93M diet, while those on the HFD showed an increased trend in the CUMS group, and mice on the SUP diet were unaffected.

Analysis of hippocampal SERT levels showed a significant main effect of CUMS (*p* < 0.0001) and significant increases relative to the respective control groups in the CUMS AIN93M (0.2389 ± 0.10, *p* = 0.0486), CUMS HFD (0.3918 ± 0.10, *p* = 0.0007), and CUMS SUP groups (0.3272 ± 0.10, *p* = 0.0046). These results suggest that CUMS increased hippocampal SERT levels regardless of diet.

In the PFC, two-way ANOVA of TPH2 levels revealed a significant main effect of CUMS (*p* = 0.0269) and a significant main effect of diet (*p* = 0.0099, main effect; *p* = 0.0114, AIN93M vs. HFD; *p* = 0.0663, AIN93M vs. SUP). A significant increase in TPH2 levels was found in the HFD control compared with the AIN93M control group (0.2965 ± 0.10, *p* = 0.0116). These results suggest that CUMS treatment increased TPH2 levels in the PFC, while the HFD also contributed to increased TPH2 levels.

Regarding IDO, a main effect of diet (*p* = 0.0252, main effect; *p* = 0.0218, HFD vs. SUP) and an interaction effect was found (*p* = 0.0029). Comparing the control groups, a significant increase was found in the HFD control vs. AIN93M control groups (0.1015 ± 0.03, *p* = 0.0144). CUMS treatment increased PFC IDO levels in mice on the AIN93M diet (0.0945 ± 0.03, *p* = 0.0245), however, IDO levels in the CUMS SUP group were significantly lower than both the CUMS AIN93M group (−0.1158 ± 0.03) and the CUMS HFD group (−0.0910 ± 0.03, *p* = 0.0317). A near-significant decrease was also found in the CUMS SUP group compared with the control SUP (−0.0780 ± 0.03, *p* = 0.0786). These results suggest that CUMS increased IDO levels, but this effect was differentially affected by diet.

Analysis of MAO-A showed a near-significant main effect of CUMS (*p* = 0.0660) and a near-significant interaction effect (*p* = 0.0707). MAO-A levels were unaffected by CUMS in the AIN93M and HFD diets. However, a significant increase was found in the CUMS SUP group compared with the control SUP group (0.6446 ± 0.21, *p* = 0.0150). MAO-A levels in the control SUP group were also almost significantly lower than those in the control HFD group (−0.5352 ± 0.21, *p* = 0.0525). These results suggest that CUMS increased MAO-A levels in the SUP diet, although the HFD showed overall increased trends regardless of CUMS treatment.

Responsible for releasing serotonin into the synaptic cleft, VMAT2 levels in the PFC appeared unaffected by CUMS treatment in any of the diets. However, a near significant increase in VMAT2 levels was detected in the control HFD group compared with the control AIN93M group (0.4141 ± 0.17, *p* = 0.0623) as well as a significant increase in the control SUP group compared with the control AIN93M group (0.4413 ± 0.17, *p* = 0.0428).

Analysis of SERT levels in the PFC showed a significant main effect of CUMS (*p* < 0.0001) and diet (*p* < 0.0001, main effect; *p* = 0.0020, AIN93M vs. HFD; *p* < 0.0001, AIN93M vs. SUP). Compared with their respective control groups, SERT levels were increased in the CUMS AIN93M (0.4816 ± 0.12, *p* = 0.0009), CUMS HFD (0.5137 ± 0.12, *p* = 0.0004) and the CUMS SUP groups (0.8073 ± 0.12, *p* < 0.0001). SERT levels were lowest in the control AIN93M group, with increased levels in the control HFD (0.2996 ± 0.12, *p* = 0.0485) and control SUP (0.3140 ± 0.12, *p* = 0.0363) groups by comparison. SERT levels were also higher respective to the CUMS AIN93M group in the CUMS HFD (0.3317 ± 0.12, *p* = 0.0252) and CUMS SUP (0.6396 ± 0.12, *p* < 0.0001), and also between the CUMS SUP and CUMS HFD groups (0.3079 ± 0.12, *p* = 0.0411). These results suggest that CUMS increase SERT levels in the PFC, with this effect exacerbated in the HFD and SUP diets.

### 2.5. Mature Brain-Derived Neurotrophic Factor Levels in the Hippocampus and Prefrontal Cortex

Mature brain-derived neurotrophic factor (mBDNF) levels, which promote neuroplasticity, were also evaluated in the hippocampus and PFC (Figure 8). In the hippocampus, mBDNF levels were unaffected by both CUMS and diet. In the PFC, a significant main effect of diet was found (*p* = 0.0126, main effect; *p* = 0.0110, AIN93M vs. SUP) as well as a significant increase in mBDNF levels in the CUMS SUP group compared to the control SUP group (0.3490 ± 0.14, *p* = 0.0454), with a similar increased trend in the control SUP compared with the control AIN93M, suggesting that the SUP diet increased mBDNF levels in the PFC.

### 2.6. Serotonin Synthesis and Metabolism Pathways in the Colon

Colonic serotonin synthesis and metabolism pathways were evaluated by western blotting, and the results are presented in Figure 9. TPH2, mainly expressed in neurons, showed a significant main effect of diet (*p* = 0.0633), with decreased trends in the HFD mice versus the SUP mice (*p* = 0.0595). Tryptophan hydroxylase 1 (TPH1), mainly expressed in enterochromaffin cells of the colon, showed a significant main effect of diet (*p* = 0.0052, main effect; *p* = 0.0479, AIN93M vs. HFD; *p* = 0.0056, HFD vs. SUP). TPH1 levels were increased in the CUMS SUP group compared with the CUMS AIN93M (0.1680 ± 0.15, *p* = 0.0586) and CUMS HFD groups (0.3075 ± 0.15, *p* = 0.0319). Regarding serotonin metabolism, analysis of colonic IDO levels showed a significant main effect of diet (*p* = 0.0204, main effect; *p* = 0.0194, HFD vs. SUP) and a near significant main effect of CUMS (*p* = 0.0710), suggesting that CUMS increased IDO levels. Analysis of MAO-A levels showed a significant main effect of diet (*p* = 0.0083, main effect; *p* = 0.0071, HFD vs. SUP), with a decrease in levels in the CUMS SUP group compared with the CUMS HFD group (0.6962 ± 0.27, *p* = 0.0497). Meanwhile, SERT levels showed a significant main effect of CUMS treatment (*p* = 0.0025), with a significant decrease in CUMS HFD compared with the control HFD (−0.1348 ± 0.22, *p* = 0.0044).

Taken together, these results suggest that while colonic serotonin production was likely to be increased in the SUP diet, CUMS treatment, as well as the SUP diet, increased the metabolism of serotonin via the kynurenine pathway, while CUMS may have decreased serotonin reuptake in the HFD. However, serotonin metabolism via MAO-A was reduced in the SUP diet.

### 2.7. Colonic Histopathology

Since dietary effects were identified in colonic serotonergic regulation, the histology of the colon was investigated (Figure 10). Two-way ANOVA of the measurements of the colonic crypt length showed a significant main effect of diet (*p* = 0.0036, main effect; *p* = 0.0062, AIN93M vs. SUP; *p* = 0.0125, HFD vs. SUP), with crypt lengths significantly increased in the CUMS SUP group compared with the CUMS AIN93M group (50.65 ± 14.29, *p* = 0.0121) and with the CUMS HFD group (55.69 ± 14.29, *p* = 0.0064), while the control SUP group had an increased trend in comparison with the AIN93M and HFD controls with no significant difference between the control and CUMS SUP groups. This suggests that the SUP diet increased the colonic crypt length but that four weeks of CUMS treatment did not affect crypt length.

A cell count of the goblet cells within colonic crypts was carried out, and two-way analysis of the results showed a significant main effect of diet (*p* = 0.0009, main effect; *p* = 0.0037, AIN93M vs. SUP; *p* = 0.0014, HFD vs. SUP) and a significant main effect of CUMS (*p* = 0.0003). CUMS treatment reduced the number of goblet cells in the AIN93M (−68.67 ± 21.95, *p* = 0.0261) and HFD (−69.67 ± 21.95, *p* = 0.0240) groups compared to their respective controls. The SUP diet attenuated these effects since no significant difference was found between the SUP control and CUMS groups (*p* = 0.0828). This was corroborated by the higher number of goblet cells in the CUMS SUP group compared with the CUMS AIN93M (72.00 ± 21.95, *p* = 0.0197) and CUMS HFD (81.00 ± 21.95, *p* = 0.0093) groups. The number of goblet cells was also increased in the control SUP group compared with the control AIN93M (58.33 ± 21.95, *p* = 0.0626) and control HFD (66.33 ± 21.95, *p* = 0.0318) groups. The results suggest CUMS treatment reduced the number of goblet cells, while the SUP diet increased the number of goblet cells and alleviated the reduction in goblet cell numbers brought about by CUMS treatment.

### 2.8. Gut Microbiota

The diversity and composition of the gut microbiota was evaluated in faecal pellet samples. The species richness and evenness of the gut microbiota within samples across the groups were measured using the number of observed operational taxonomic units (OTUs) and the Gini-Simpson Index (Figure 11). Regarding the richness of the gut microbiota, the CUMS AIN93M group had a decreased trend in the number of OTUs compared with the control AIN93M group, while the groups for the HFD and SUP remained unchanged, suggesting that CUMS may have mildly reduced the richness of the gut microbiota in mice on the AIN93M diet.

Analysis of the Simpson Index revealed a main effect of diet (*p* = 0.0013, main effect; *p* = 0.0083, AIN93M vs. SUP; *p* = 0.0018, HFD vs. SUP) and a significant decrease in the Simpson Index in the SUP diet compared with the AIN93M diet for the control groups (−0.1880 ± 0.07, *p* = 0.0487) and a near significant decrease in the CUMS groups (0.1693 ± 0.07, *p* = 0.0814). The Simpson Index in the control SUP group was also lower to an even greater extent than the control HFD group (0.2960 ± 0.07, *p* = 0.0026). Since a higher value on the index measured indicates less even distribution of species, the results suggest that the SUP diet increased the evenness of the gut microbiota in both control and CUMS mice.

The dissimilarities in microbial populations between samples were assessed using the Bray-Curtis β-Diversity index. Pairwise PERMANOVA comparisons (Table 1) showed near-significant dissimilarities between the control AIN93M group and the control HFD group (*p* = 0.078) and between the control HFD and control SUP groups (*p* = 0.089), suggesting that the gut microbiota of mice on the HFD was slightly different to that of the AIN93M and SUP diets which more closely resembled each other. A near-significant dissimilarity between microbial populations was also found in the CUMS AIN93M group compared with the CUMS SUP group (*p* = 0.078) and between the CUMS HFD and CUMS SUP groups (*p* = 0.088). These results, although mild, suggest that while the microbial populations in AIN93M and SUP diets in controls more closely resembled each other, CUMS increased the dissimilarities between them. CUMS did not affect the dissimilarity between the HFD and SUP groups. However, the microbial populations were not more dissimilar from the control counterparts in any of the diets.

Following analyses of the diversity of the gut microbiota, the relative abundances of dominant bacterial phyla were analysed to identify major shifts in the gut microbiota. The bacterial phyla, which consisted of greater than 1% of the total microbiota, were Actinobacteria, Bacteroidetes, Firmicutes and Proteobacteria and are presented in Figure 12. The relative abundance of Actinobacteria was affected by diet (*p* < 0.0001, main effect; *p* = 0.0096, AIN93M vs. HFD; *p* < 0.0001, AIN93M vs. SUP; *p* < 0.0001, HFD vs. SUP) and an interaction effect was also found (*p* = 0.0030). Actinobacteria abundance was increased in the control SUP group relative to the control AIN93M group (62.05 ± 7.18, *p* < 0.0001), while the control SUP group had significantly higher Actinobacteria abundance than the control HFD group (60.00 ± 7.18, *p* < 0.0001). Compared with the CUMS AIN93M group, Actinobacteria abundance was lower in the CUMS HFD group (−39.30 ± 7.18, *p* = 0.0004) and higher in the CUMS SUP group (26.26 ± 7.18, *p* = 0.0098), while the CUMS HFD group had lower abundance compared with the CUMS SUP group (−65.56 ± 7.18, *p* < 0.0001). Comparing CUMS versus control groups, CUMS increased Actinobacteria abundance in the AIN93M diet (27.91 ± 7.18, *p* = 0.0065), and appeared to decrease abundance in the HFD, although this did not reach significance. Meanwhile, CUMS had no effect on Actinobacteria abundance in the SUP diet.

Analysis of the Bacteroidetes abundance showed a significant main effect of diet (*p* = 0.0375, main effect; *p* = 0.0457, HFD vs. SUP) and a near significant interaction effect (*p* = 0.0375). A significant increase in Bacteroidetes abundance was found in the CUMS SUP group compared with the CUMS HFD group (10.08 ± 3.26, *p* = 0.0279).

Firmicutes abundance showed a significant main effect of diet (*p* < 0.0001, main effect; *p* = 0.0005, AIN93M vs. HFD; *p* < 0.0001, AIN93M vs. SUP; *p* < 0.0001, HFD vs. SUP) and an interaction effect (*p* = 0.0033). Amongst the controls, Firmicutes abundance was reduced in the SUP group compared with the AIN93M group (−54.42 ± 5.83, *p* < 0.0001) and compared with the HFD group (−58.53 ± 5.83, *p* < 0.0001). CUMS reduced Firmicutes abundance in the AIN93M diet in comparison with controls (−18.07 ± 5.83, *p* = 0.0276), while in the HFD CUMS increased Firmicutes abundance relative to controls (17.82 ± 5.83, *p* = 0.0299). A significant increase was also found between the CUMS HFD and CUMS AIN93M groups (39.99 ± 5.83, *p* < 0.0001), while the CUMS HFD group also had greater abundance compared with the CUMS SUP group (73.88 ± 5.83, *p* < 0.0001). The CUMS SUP group also had significantly lower Firmicutes abundance than the CUMS AIN93M group (−33.89 ± 5.83, *p* = 0.0002).

Since major shifts in the gut microbiota composition were identified at the phylum level, the changes were further investigated by evaluating the microbial composition at the genus level (Figure 13 and Figure 14). Two-way ANOVA of the relative abundance of Lactobacillus, belonging to the Firmicutes phylum, revealed a main effect of diet (*p* = 0.0169, main effect; *p* = 0.0195, HFD vs. SUP) and a significant decrease in the CUMS SUP group compared with the CUMS HFD group (38.02 ± 13.50, *p* = 0.0467). Analysis of Faecalibaculum, another member of the Firmicutes phylum, showed a significant main effect of diet (*p* = 0.0002, main effect; *p* = 0.0002, AIN93M vs. SUP; *p* = 0.0038, HFD vs. SUP) and an interaction effect (*p* = 0.0062). CUMS treatment significantly reduced Faecalibaculum abundance in the AIN93M diet (−30.03 ± 7.94, *p* = 0.0078), while CUMS did not affect the HFD and SUP diets. A significant decrease was also found in the CUMS SUP group compared with the CUMS HFD group (−30.43 ± 7.94, *p* = 0.0071). In the control groups, Faecalibaculum abundance was reduced in the control HFD group compared to the control AIN93M group (32.23 ± 7.94, *p* = 0.0047), and in the control SUP group compared with the control AIN93M group (48.74 ± 7.94, *p* = 0.0002). These results suggest that Lactobacillus and Faecalibaculum abundance was drastically reduced in the SUP diet and may have contributed to the changes in abundance in the Firmicutes phylum.

Belonging to Actinobacteria, Bifidobacterium abundance had a significant main effect on a diet (*p* < 0.0001, main effect; *p* = 0.0006, AIN93M vs. HFD; *p* < 0.0001, AIN93M vs. SUP; *p* < 0.0001, HFD vs. SUP), a near significant main effect of CUMS (*p* = 0.0722) and a significant interaction effect (*p* = 0.0043). While CUMS did not affect the HFD and SUP diets, it significantly increased Bifidobacterium abundance in the AIN93M diet (32.69 ± 7.23, *p* = 0.0021). Regarding the effects of diet, the SUP diet had the highest abundance of Bifidobacterium, with significantly higher abundance in the control SUP group compared with the control AIN93M (78.80 ± 7.23, *p* < 0.0001) and with the control HFD (89.62 ± 7.23, *p* < 0.0001) groups. Higher abundance was also found in the CUMS SUP group compared with the CUMS AIN93M (38.12 ± 7.23, *p* = 0.0006) and CUMS HFD (81.62 ± 7.23, *p* < 0.0001) groups. Another member of the Actinobacteria phylum, Atopobiaceae (uncultured bacterium), showed a significant main effect of diet (*p* = 0.0155, main effect; *p* = 0.0291, AIN93M vs. SUP; *p* = 0.0376, HFD vs. SUP), with the abundance significantly reduced in the control SUP group in comparison with the control HFD group (−19.55 ± 6.35, *p* = 0.0288). Analysis of Enterorhabdus abundance revealed a near-significant main effect of diet (*p* = 0.0650, main effect). However, no specific differences were found between the groups.

Members of the Bacteroidetes phylum, such as Bacteroides, showed a significant main effect of diet too (*p* = 0.0453; *p* = 0.0456, HFD vs. SUP), and a near-significant increase in abundance in the CUMS SUP group compared with the CUMS HFD group (1.830 ± 0.69, *p* = 0.0634). Muribaculaceae (uncultured bacterium) abundance had a near-significant main effect of diet (*p* = 0.0640, main effect), an interaction effect (*p* = 0.0687), and a near-significant increase in the CUMS SUP group compared with the CUMS HFD group (6.089 ± 2.30, *p* = 0.0642).

### 2.9. Adrenal Catecholamine Synthetic Enzymes and SERT

The regulation of adrenal catecholamines was also evaluated by western blotting (Figure 15). Two-way ANOVA of total TH levels in the adrenal gland showed no effects of diet or CUMS treatment. The phosphorylation of adrenal TH was then investigated by evaluating the phosphorylation levels of the sites of serine residue 19 (pSer19TH), 31 (pSer31TH) and 40 (pSer40TH) on TH. Analysis of pSer19TH showed a significant main effect of CUMS treatment (*p* < 0.0001), with a reduction in pSer19TH levels in the CUMS AIN93M versus control AIN93M group (−0.4568 ± 0.14, *p* = 0.0070), CUMS HFD versus control HFD group (−0.4751 ± 0.14, *p* = 0.0049), and in the CUMS SUP versus control SUP group (−0.3650 ± 0.14, *p* = 0.0374), suggesting that CUMS reduced the phosphorylation of TH at serine residue 19. For pSer31TH, a significant main effect of CUMS treatment (*p* = 0.0008) and an interaction effect (*p* = 0.0201) was found. CUMS treatment reduced pSer31TH levels in the AIN93M diet compared with the respective control group (−0.6847 ± 0.18, *p* = 0.023), while levels were similarly reduced in the HFD CUMS and control groups (−0.5386 ± 0.18, *p* = 0.0185). Interestingly, the control SUP group had a reduction in pSer31TH compared with the control AIN93M group (−0.5071 ± 0.18, *p* = 0.0283) and no difference between the respective CUMS group. For pSer40TH, a significant main effect of CUMS treatment (*p* = 0.0035) and a near significant main effect of diet (*p* = 0.0775, main effect) were found. CUMS treatment reduced pSer40TH levels in the CUMS AIN93M group compared with the control AIN93M group (−0.5266 ± 0.18, *p* = 0.0231), while a decreased trend appeared in the HFD CUMS group compared with the control HFD. No differences were found in the CUMS SUP group compared to the control group. However, the control SUP group had significantly reduced pSer40TH levels compared with the control AIN93M (−0.5097 ± 0.18, *p* = 0.0289) and control HFD (−0.4814 ± 0.18, *p* = 0.0405) groups. These results suggest that pSer31TH and pSer40TH CUMS treatment reduced phosphorylation in the AIN93M and HFD diets. However, levels were lower in the SUP diet and unaffected by CUMS treatment.

Next, adrenal phenylethanolamine N-methyltransferase (PNMT) and SERT levels were investigated. However, no effects of CUMS treatment or diet were found for PNMT. For SERT, a significant main effect of diet was found (*p* < 0.0001, main effect; *p* < 0.0001, AIN93M vs. SUP; *p* < 0.0001, HFD vs. SUP). Considering the diets, a significant increase in adrenal SERT levels was found in the control SUP group compared with the control AIN93M (1.593 ± 0.34, *p* = 0.0002) and control HFD (1.899 ± 0.34, *p* < 0.0001) groups. A significant increase was also found in the CUMS SUP group compared with the CUMS AIN93M (1.008 ± 0.34, *p* = 0.0191) and CUMS HFD (0.9280 ± 0.34, *p* = 0.0340) groups, suggesting that CUMS did not affect adrenal SERT levels, but that the SUP diet increased levels compared to the AIN93M and HFD which showed similar levels.

## 3. Discussion

### 3.1. General Biochemistry

While CUMS did not affect body weight or general biochemistry, the different diets used in this study did. Relative to the AIN93M diet, the mice on the HFD had a higher weight, and those on the SUP diet had the lowest weight. Serum triglycerides and cholesterol were lower in the SUP diet compared to both the AIN93M and HFD, while serum cholesterol levels in the HFD were higher than those in the AIN93M diet and SUP diet, as expected. Since the AIN93M diet was formulated to maintain the long-term health of rodents, the decrease in cholesterol levels in mice on the SUP diet relative to the AIN93M diet was surprising since this diet was based on the AIN93M diet. Such low levels of cholesterol in mice on the SUP diet may have been due to the amount of viscous soluble fibre in the diet, largely due to the inclusion of psyllium husk since such fibres impede cholesterol absorption by enterocytes through the formation of a viscous matrix in the gut lumen [39]. It was recently shown that psyllium husk administered to C57Bl6 mice reversed high-fat diet-induced hypercholesterolemia and reduced obesity. However, our study shows that the inclusion of psyllium husk into the diet of mice of normal body weight may not be beneficial to general health since their cholesterol levels were below the healthy reference range for C57Bl6 mice [40,41].

### 3.2. Behaviour

The CUMS model has been well established in the literature as an effective rodent model to induce anxiety-like behaviours in the OFT and EPM test and depression-like behaviours in the TST and forced swim test [42,43,44]. In the present study, we show that CUMS induced anxiety-like behaviour in the OFT but not the EPM test. While most studies use standard chow as a control diet, many do not account for the experimental variability that may be brought about through non-purified diets [37,45]. Moreover, previous studies have highlighted that, in chronic stress models, mice fed non-purified diets show differences in behavioural outcomes from those fed purified diets [38,46]. Thus, interpreting behavioral results from studies using purified diets with those published using non-purified diets requires caution and comparisons should be drawn between other purified diet studies. The increased anxiety-like behaviour in the OFT induced by CUMS in our study corroborates the results reported in a previous study which showed increased anxiety-like behaviour following a 5-week CUMS protocol on an AIN93M diet [47]. Liu et al. [47] also showed that CUMS increased anxiety-like behaviour in the EPM by decreasing open-arm entries. The present study showed that mice on the AIN93M diet did not exhibit this decrease in open-arm entries compared to controls. While Liu et al. [47] used a similar protocol in their study, this discrepancy may have been due to the shorter duration of the CUMS protocol used in our research and since the 24-h food and water deprivation was omitted from our protocol for animal welfare.

In the HFD used in this study, mice in neither the control nor the CUMS group showed anxiety-like behaviour even though previous studies show that high-fat diets increase anxiety-like behaviour [48,49]. This may have been due to the HFD in our study being a 23% fat, high glycaemic index diet rather than a more traditional HFD where only fat content is increased. Moreira Júnior et al. [50] showed that a high-sugar, high-fat diet had anxiolytic effects on behaviour in C57Bl6 mice compared with controls fed an AIN93G diet. They suggested that consumption of such high GI diets provides relief from negative emotional states through the activation of reward pathways in the brain and induces compulsive behaviour.

The present study also found locomotion was affected in both the OFT and EPM in mice on the SUP compared with those on the AIN93M diet. Thus, the results for entries into the zones and arms in the OFT and EPM, respectively, were corrected for the total distance travelled by each mouse to resolve better whether the differences found for each resulted from locomotion alone or anxiety-like behaviour. Following correction, the CUMS SUP mice showed a near significant decrease in central zone entries compared with their control counterparts in the OFT. The number of closed-arm entries in the EPM also increased because of CUMS in the SUP diet, suggesting increased anxiety-like behaviour. Furthermore, analyses of the decreased duration of time spent in the open arms and the increased time in the closed arms of EPM in mice on the SUP diet also suggest that the SUP diet increased anxiety-like behaviour. This surprising behaviour may have been due to the presence of psyllium husk, an indigestible resistant starch, in the SUP diet since it has been previously shown that supplementation of diets rich in resistant starches increases anxiety-like behaviour in mice [51], even though studies which administered high fibre, fruit and vegetable powder modifications of the AIN93M diet are lacking. Furthermore, the increased anxiety-like behaviour in mice on this diet may also have been related to abnormally low cholesterol levels since lower cholesterol levels have been associated with anxiety [52].

Regarding the TST, mice on the AIN93M diet did not show significantly different durations of immobility time. However, mice exposed to the CUMS protocol tended to have slightly longer periods of immobility which may suggest a propensity towards depressive-like behaviour, since CUMS has been well-established to increase immobility in the TST [43,53]. In the HFD, mice showed slightly longer durations of immobility compared to control AIN93M mice. However, this was not significantly different, and CUMS did not affect immobility in mice on the HFD, suggesting that, like anxiety, the palatability of the HFD provided relief from depressive-like behaviour. This may have been due to the amount of fat in the diet (23% or 42 kJ%) since a study by Yu et al. [54] showed that 60 kJ% high-fat diet did not protect against a stress-induced increase in immobility time in the TST.

Interestingly, mice on the SUP diet had reduced immobility time overall, but especially in the CUMS SUP group compared with the CUMS AIN93M group, suggesting that the SUP diet may have had an anti-depressive effect. Previous studies have shown that supplementation with components of fruits such as berries and vegetables such as spinach containing amino acids, essential vitamins, flavonoids, polyphenols, and folic acid reduced immobility time in the TST [55,56,57,58]. With such unexpected results in general biochemistry and behaviour, serotonin pathways in the hippocampus and PFC were investigated to further elucidate the effects of each diet.

### 3.3. Serotonin Synthesis and Metabolism Pathways in the Brain

In a normally functioning brain, serotonergic pathways originating from the dorsal raphe nucleus in the brain stem project to the hippocampus to modulate learning and memory and project to the medial PFC to modulate behavioural impulsivity [59]. Serotonin production is rate-limited in presynaptic neurons by TPH2, which converts dietary tryptophan into 5-hydroxytryptophan, then converted into serotonin by aromatic L-amino acid decarboxylase [59]. Previous studies have shown that CUMS decreases TPH2 protein levels in the hippocampus and PFC in rodents fed standard chow [9,10,60]. Here, we show that mice fed a semi-pure AIN93M diet subjected to a 4-week CUMS protocol showed increased hippocampal TPH2 levels. While it is clear that chronic stress and depressive states deplete forebrain regions of serotonin, the exact mechanisms are still being elucidated. Donner et al. [61] have proposed that anxiety-like responses are mediated by an exaggerated release of serotonin in forebrain regions due to a desensitisation of serotonergic autoreceptors in the dorsal raphe nucleus. According to Garcia-Garcia et al. [62], the amount of serotonin signalling through serotonergic postsynaptic heteroreceptors exerts a modulatory effect on the PFC and hippocampus during developmental periods, lasting into adolescence, and they suggest that enhancement of such signalling may enable improved coping with stressors and hedonic drive. The present study warrants future studies of the effects of the AIN93M diet on these serotonergic autoreceptors and heteroreceptors since 4-weeks of CUMS could not induce behavioural despair (depression-like behaviour) in mice on the AIN93M diet but did result in an anxiety-like phenotype, suggesting improved stress coping in these mice due to the increased TPH2 levels and lack of depressive-like behaviour in the TST. Interestingly, the HFD modification of the AIN93M diet did not show the same effect, while the SUP diet improved stress coping overall, further suggesting an influence of diet on these mechanisms. Indeed, this was reflected in hippocampal TPH2 levels since a significant increase in TPH2 levels was found in the SUP diet overall, while those in both HFD groups were unchanged relative to AIN93M controls.

Nonetheless, forebrain serotonin levels are also mediated through several other mechanisms, such as reuptake by presynaptic neurons, the kynurenine pathway and serotonin metabolism. Reuptake by presynaptic neurons is mediated by SERT, and previous studies have shown that CUMS and other chronic stress models increase SERT in the hippocampus and PFC, resulting in a reduction of serotonin in the synaptic cleft [10,11]. The present study similarly showed SERT levels were increased in CUMS in the hippocampus. This suggests a reduction in serotonin availability for postsynaptic neurons in the hippocampus because of serotonin reuptake. Interestingly, no changes in VMAT2 levels were found in mice exposed to CUMS treatment or as an effect of any diet. VMAT2 is responsible for the release of cytoplasmic serotonin into the synaptic cleft [59]. Thus, these results suggest that serotonergic neurotransmission was indeed decreased in the hippocampus because of increased reuptake.

With increased hippocampal TPH2 and SERT levels suggesting increased cytoplasmic serotonin levels, serotonin metabolism was investigated. MAO-A, responsible for the metabolism of serotonin into 5-hydroxyindoleacetic acid and whose expression has previously been shown to be decreased in CUMS was similarly decreased overall by CUMS in the present study and unaffected by diet [8]. Finally, the rate-limiting enzyme of the kynurenine pathway, IDO, was investigated. Readers are referred to the reviews by Savitz [16] and Schwarcz and Stone [63] for a detailed discussion of the kynurenine pathway. Previous studies have consistently reported increased IDO levels in the hippocampus of rodents subjected to CUMS and fed standard chow diets [8,60,64]. Here we show that mice subjected to CUMS on semi-pure diets show a similar increase in hippocampal IDO, suggesting up-regulation of the kynurenine pathway, which may increase neurotoxicity [16]. However, this effect was least obvious in mice on the SUP diet.

In the PFC, TPH2 levels were similarly increased by CUMS, suggestive of top-down control of stress coping [62]. However, a dietary effect was also found with increased levels in the HFD and SUP diets. These dietary effects may have been due to the fatty acid compositions in each diet since higher amounts of ω-6 fatty acids are pro-inflammatory and increase insulin secretion [65]. Such increases in insulin are also known to increase plasma tryptophan levels, which contribute to increased serotonin levels in the brain [66]. Considering the composition of the diets (see Appendix A), the HFD had the highest levels of ω-6 fatty acids. The SUP diet had lower levels of ω-6 fatty acids than the AIN93M diet, although the AIN93M diet had slightly higher anti-inflammatory ω-3 fatty acids than the SUP diet. Thus, the SUP diet may have increased plasma tryptophan levels more than the AIN93M diet, even though the SUP diet contained the least amount of tryptophan.

Serotonin reuptake by SERT in the PFC was also increased in mice exposed to CUMS. However, an effect of diet was also found in this brain region, with the HFD and SUP diets having higher levels than the AIN93M diet, even in controls, with levels highest in the SUP diet. The CUMS protocol is known to induce neuroinflammation [67], and according to Ghaffari-Nasab et al. [68], higher levels of inflammation in the brain are associated with increased SERT expression, particularly in the PFC, reducing serotonergic function. Thus, the increased SERT levels in the PFC of mice in the CUMS group of each diet may have been due to CUMS-induced neuroinflammation. However, the increase in SERT levels in the control mice in the HFD and SUP diet compared with the AIN93M mice may not have been related to increased neuroinflammation since increased VMAT2 levels were also found in these groups, indicative of increased release of serotonin into the synapse, suggesting that there was no net effect of the increased SERT levels in these mice. However, mice in the CUMS group of each diet did not exhibit changes to VMAT2 levels. Thus, the increased SERT in these groups would have reduced extracellular serotonin levels.

Regarding the intracellular levels of serotonin in the PFC, MAO-A levels were only significantly increased in mice exposed to CUMS on the SUP diet compared to controls, while those in the HFD control group had higher levels than SUP controls suggesting increased metabolism of serotonin in these groups. Interestingly, an interaction effect between diet and stress on PFC IDO levels was found with increased IDO levels in the AIN93M CUMS mice compared with controls and decreased levels in SUP CUMS mice compared with controls, while mice on the HFD had higher IDO levels overall regardless of exposure to stress. Increased IDO levels have been associated with increased inflammation [63]. Thus, the increased levels in stressed mice on the AIN93M diet may be another consequence of neuroinflammation. Diets high in fibre are thought to reduce neuroinflammation since they have been associated with lower levels of TNF- and IL-6 [69,70]. Thus, the findings for PFC SERT levels in the SUP mice are less suggestive of increased neuroinflammation due to the reduction seen in IDO levels. Other mechanisms, therefore, likely contributed to an increase in SERT in the PFC of CUMS SUP mice.

### 3.4. Mature Brain-Derived Neurotrophic Factor Levels in the Hippocampus and Prefrontal Cortex

Previous studies have shown that CUMS results in a decrease in hippocampal and PFC levels of the neurotrophin BDNF in association with depressive-like behaviour [53,71,72]. The present study found CUMS in mice fed semi-pure diets resulted in no alterations to hippocampal or PFC BDNF levels. However, the SUP diet appeared to increase BDNF levels overall, suggesting a beneficial effect of this diet on neuroplasticity. This increase may have also been related to the increase in PFC SERT levels in mice on this diet since Diniz et al. [73] previously showed a relationship between PFC BDNF and SERT since SERT knockout mice exhibit decreased BDNF levels and depressive-like behaviour and downregulation of BDNF mRNA in the PFC and hippocampus. Another explanation for the augmentation of BDNF in the CUMS SUP mice may be that the downregulation of the kynurenine pathway seen as a decrease in IDO protein levels may have reduced the levels of neurotoxic metabolite of the kynurenine pathway, quinolinic acid. According to Correia and Vale [5], BDNF levels are reduced, and glial-neuronal networks are weakened by quinolinic acid thus the reduction in IDO may have been responsible for the increase in BDNF in these mice via a reduction in quinolinic acid in the PFC. Thus, the SUP diet may have exerted the reduction in behavioural despair in the TST through an increase in PFC BDNF levels and, although this increased SERT levels, the increased TPH2 and MAO-A levels suggest that there was no net change in PFC serotonin levels because of CUMS.

### 3.5. Colonic Serotonin Synthesis and Metabolism Pathways, Histopathology, and the Gut Microbiota

Serotonin production is not limited to the central nervous system. 95% of serotonin production from dietary tryptophan occurs in the gut [8,74]. Li et al. [8] previously showed that CUMS induces substantial alterations to tryptophan metabolism in the cortex, hippocampus, and colon in rats, with increased IDO expression and kynurenine content in all three tissues. Importantly, 3-hydroxycaninuric acid was increased in the brain regions in their study. At the same time, the colon showed an increase in kynurenic acid content, suggesting that the neurotoxic branch of the kynurenine pathway is favoured in the brain.

In contrast, the neuroprotective branch is favoured in the gut in CUMS rodents [8]. In the present study, IDO protein levels in the colon were increased overall in the SUP diet, while a near-significant overall effect of CUMS was found. According to Koopman et al. [74], increased colonic serotonin can also contribute to intestinal inflammation via serotonin receptor activation on dendritic cells, macrophages and T cells, triggering pro-inflammatory pathways. Colonic SERT controls excess serotonin through reuptake into enterocytes where it is metabolized by MAO-A. The present study showed CUMS decreased colonic SERT levels, exacerbated by the HFD. While MAO-A levels were not affected by CUMS, suggesting CUMS increased colonic serotonin levels, which may have promoted pro-inflammatory pathways, which was most apparent in the HFD.

Increased colonic inflammation has also been associated with disruption of intestinal barrier activity and the mucosal barrier since it was shown that CUMS in rats increased colonic IFN-γ and IL-6, and decreased tight junction protein expression and goblet cell numbers [75]. Our study similarly showed CUMS induced a significant reduction in goblet cell numbers in the AIN93M diet and HFD, but this effect did not occur in the SUP diet. This effect was likely due to the high-fibre content of the SUP diet since mice on a low-fibre diet show increased bacterial degradation of the mucus barrier. Thus, a high-fibre diet may promote goblet cell function [76]. Indeed, the colonic crypt length was increased only in the SUP diet, likely to accommodate increased goblet cell expression.

The mucus barrier of the colonic epithelium is of high importance to the gut microbiota since it provides a layer for colonization, but also since it is an energy source for some commensal bacteria which release short-chain fatty acids as a by-product, further enhancing goblet cell mucus production [74,77]. Many recent studies have reported the effects of various chronic stress models on the diversity and composition of the gut microbiota. However, no consensus has been established due to variability in the alterations reported (readers are referred to the previous review [78]). Such variability may be due to differences in methodologies used across different studies, but it is also plausible that non-purified standard chow may also be a contributing factor. The present study reports that four weeks of CUMS in mice on a semi-pure AIN93M diet did not affect the diversity of the gut microbiota, although a decreased trend in microbial richness was found. High-fat, high-GI and high-fibre “superfood” modifications of the AIN93M diet did not show any effects of CUMS on the richness or evenness of the microbial diversity. However, the SUP diet did increase the evenness of the gut microbiota overall.

Compositionally, at the phylum level, a healthy gut microbiota is considered to have a low Firmicutes to Bacteroidetes ratio [79]. This ratio was unchanged by CUMS in mice on all three diets (see Appendix A, however, mice on the HFD showed a higher abundance of Firmicutes and a lower abundance of Bacteroidetes compared to the AIN93M and SUP diets, suggesting that the HFD established a less healthy gut microbiota composition. Previous studies also showed that CUMS increases Actinobacteria abundance similarly to depressive patients [17,18], which the present study corroborated, finding that mice in the CUMS AIN93M group had higher levels than controls. Interestingly, while CUMS showed no effect on Actinobacteria abundance in mice on the SUP diet, the abundance of this phylum was increased overall in this diet. This may be suggestive of a “depressive” gut-microbiota phenotype. However, translational comparisons of such results to humans require caution as the rodent gut microbiota may not be entirely comparable to that of humans since a considerable proportion of microbes of the human gut are unable to colonise the rodent gut due to the absence of human genotypic and lifestyle factors [80]. Furthermore, the behavioural results in the present study do not support the proposition that Actinobacteria plays a role in a depressive phenotype in mice.

Greater variability exists in the reported effects of chronic stress on the genera of the gut microbiota. A recent study by Zhang et al. [79] compared the composition of the gut microbiota of mice in a CUMS model to those in a maternal separation model and found they were similar compositionally. However, they differed significantly in some specific relative abundances. The study concluded that the gut microbiome can be affected differently depending on the type, duration and intensity of the stressors used to model depression [79]. Nonetheless, studying the host-microbiome interactions in chronic stress models may yield important findings. It was previously shown that SERT knockout mice have increased Lactobacillus and decreased Bifidobacteria abundance associated with intestinal inflammation [74]. Bifidobacteria also increase mucus production by goblet cells in the intestinal epithelium [77]. In the present study, Lactobacillus abundance showed an increased trend in the CUMS AIN93M group and the HFD overall, while its abundance was significantly decreased in the SUP diet.

Meanwhile, the SUP diet remarkably increased Bifidobacterium abundance and moderately so in the AIN93M CUMS mice, while it was not detected in the HFD. These results may suggest that Bifidobacterium abundance contributed to the increase in goblet cell numbers in the SUP diet. In contrast, the increase in Lactobacillus abundance in the CUMS AIN93M group may have prevented this same effect, resulting in a reduction in goblet cell numbers. However, further studies are needed to investigate this association.

Other notable findings in the present study are those of the Faecalibaculum and Bacteroides abundance. It has previously been suggested that purine metabolism is dysregulated in depressive patients [81], while a recent preclinical study found that xanthine oxidase, a key enzyme in purine metabolism, is decreased in the cerebral cortex in rodents in association with anxiety- and depression-like behaviours [82]. The same study also reported that Faecalibaculum abundance was negatively correlated with xanthine oxidase activity levels, suggesting that Faecalibaculum may contribute to anxiety- and depression-like behaviours [82]. Yang et al. [83] reported that in a high-fat diet-induced obesity model, anxiety-like behaviour was associated with lower Faecalibaculum abundance. In our study, Faecalibaculum was decreased by CUMS in the AIN93M diet, while the HFD showed decreased trends but no effect of CUMS. The SUP diet showed significantly lower Faecalibaculum abundance overall, and these findings were reflected in the measures of anxiety-like behaviour, with the SUP mice displaying higher levels of anxiety-like behaviour.

It is possible that the absence of depressive-like behaviour, despite increased anxiety-like behaviour, in the mice on the SUP diet and lower immobility time in the TST overall was attributed to the abundance of Bacteroides since it was increased only in the SUP diet. Bacteroides has previously been shown to be decreased in mice exposed to CUMS [15,79]. Zhang et al. [19] also showed that Bacteroides abundance is restored in CUMS rats treated with either of the common antidepressants’ fluoxetine or amitriptyline. Thus, the increased Bacteroides abundance in the SUP mice in the present study may have contributed to the reduction in immobility time in the TST compared with the other diets.

### 3.6. Adrenal Catecholamine Synthetic Enzymes and SERT

In the present study, we analysed the rate-limiting enzyme of catecholamine biosynthesis, TH, its activation via phosphorylation events, and its downstream enzyme, PNMT, responsible for the conversion of noradrenaline to adrenaline, in the adrenal medulla [1,3]. Current literature investigating TH regulation in the adrenal medulla has focused on the effects of acute stressors on TH phosphorylation [27,28]. However, the effects of chronic stressors on TH phosphorylation have yet to be elucidated. We found that mice subjected to 4 weeks of CUMS had no changes to TH or PNMT protein levels in the adrenal medulla. A study by Santana et al. [3] on the effects of CUMS on adrenal medullary function found both TH and PNMT protein levels to be increased following one week of treatment but decreased following three weeks. These results likely indicate that shorter exposure to treatment (i.e., one week) increased the synthetic capacity of the adrenal glands of these mice, while prolonged exposure (i.e., three weeks) in their model showed a decrease in this synthetic capacity. Thus, the catecholamine synthetic capacity of the adrenal glands may be dynamically regulated based on the duration of the stress protocol used and possibly the intensity of the stressors used. In the present study, four weeks of CUMS resulted in no changes to TH or PNMT levels.

The activity of TH is regulated by almost every form of regulation, including phosphorylation at its serine residues (Ser) 19, 31 and 40 [1,84,85]. We found that CUMS decreased TH phosphorylation at the Ser19, Ser31 and Ser40 sites, with Ser31 being decreased to a greater extent than Ser40 and Ser19. According to Dunkley et al. [86], Ser31 phosphatases require the inactivation of kinases (such as ERK1/2). They are not calcium-dependent, like those for Ser19, which may be why Ser31 was dephosphorylated to a greater extent. Ser19 phosphorylation has been shown to correlate with calcium uptake in vivo, and Ca2+/calmodulin-dependent protein kinase II (CaMPKII) is thought to play a role in the phosphorylation of Ser19 [1]. Ser40 is phosphorylated by Protein Kinase A (PKA), and PKA activation has also been shown to correlate with an increase in Ser40 phosphorylation in vivo following 40 minutes of footshock stress [1].

Furthermore, there is evidence that Ser40 is dephosphorylated by protein phosphatase (PP) 2A and PP2C [86]. Therefore, the decreases in Ser19, Ser31, and Ser40 phosphorylation in our study are likely due to the downregulation of ERK1/2, CaMPKII and PKA and/or the upregulation of the phosphatases PP2A and PP2C in response to the CUMS. It is also known that phosphorylation of Ser40 and Ser31 is directly involved in the activation of TH and the subsequent release of catecholamines [85]. Our data, therefore, suggest that the decrease in phosphorylation of Ser19, Ser31 and Ser40 may cause deactivation of TH in response to CUMS, despite no changes to TH protein levels as a possible compensatory response to overstimulation of the sympathoadrenal system. Interestingly, the SUP diet decreased Ser31 and Ser40 phosphorylation by approximately 50% compared to AIN93M controls regardless of CUMS. Since patients with anxiety disorders show blunted sympathoadrenal responses as well as blunted blood pressure and heart rate reactions to acute stress [29,87], the findings in the present study suggest that the mice on the SUP diet may have had blunted sympathoadrenal responses, similar to the chronically stressed mice in the other diets, which corroborates the increased anxiety-like behaviour found in mice on this diet.

According to Brindley et al. [26], local regulation of adrenal medullary catecholamine secretion at the splanchnic-chromaffin cell synapse can occur via endocrine, auto/paracrine and neuronal transmitters. SERT is also strongly expressed in adrenal chromaffin cells, and in SERT knockout mice, restraint stress-induced catecholamine secretion is augmented [26]. Here, we showed that the SUP diet practically doubled adrenal SERT protein levels while remaining unchanged in the other diets. Thus, the increase in SERT protein levels in the SUP diet may suggest a reduction in catecholamine secretion.

### 3.7. Clinical Implications

The present study highlights the potential modulation of anxiety-like and depressive-like behaviours in mice by dietary manipulation through widespread effects on the brain, gut and sympathoadrenal medullary system. This may be relevant to clinical settings, particularly where current treatments show high relapse rates or delayed effects. Given the worldwide adoption of the “Westernised” diet, individuals may increase their risk of mental health disorders on such a diet in the long term since a diet of similar composition used in the present study showed widespread deleterious effects on the brain and gut, albeit in the absence of depressive-like and anxiety-like behaviours. Our study suggests that a high-fibre diet could exert beneficial effects on affective disorders such as depression and thus could be used as a preventative strategy or in conjunction with existing treatments to combat the adverse behavioural effects of these disorders. Importantly, however, serum cholesterol levels should be monitored in patients administered high-fibre diets to ensure healthy levels are maintained since lowered cholesterol levels may increase the risk of anxiety disorders.

### 3.8. Study Limitations

This study had limitations since depressive-like behaviour, measured as an increase in the time spent immobile in the TST, could not be produced in CUMS mice. This may have been related to the shorter duration of the CUMS protocol used since the present study used a 4-week protocol while previous studies that produced depressive-like behaviour in CUMS have used 4-week, 5-week and 6-week protocols. Furthermore, these studies commonly include food and water deprivation as a stressor in the CUMS protocol. However, this stressor was omitted in the present study regarding concerns for animal welfare since the effects of diets were an outcome measure.

The SUP diet used in this study may have exerted some negative effects due to high levels of psyllium husk, as the mice did not show normal healthy growth based on body weight throughout the study. Future studies of dose-dependent fibre supplementation in mice on semi-pure diets should formulate the diets used, ensuring the absorption and bioavailability of essential nutrients and fats remain uncompromised.

This study also did not measure free concentrations of tryptophan and its metabolites or catecholamines to confirm that any alterations in protein levels of the regulatory enzymes investigated lead to alterations in the outputs of the pathways investigated. Moreover, tryptophan content was not equilibrated across the diets, which may have contributed to the study findings related to tryptophan metabolism (see Appendix A). Quantification of the amount of diets ingested by each mouse was also not carried out. Thus, whether the amount of diet ingested influenced the results is unclear.

## 4. Materials and Methods

### 4.1. Animals

All mice were bred in the Core Animal Facility (University of South Australia), maintained under standard conditions (12:12-h light/dark cycle, lights on between 6 a.m. to 6 p.m., temperature of 22 ± 1 °C, humidity of 52 ± 2%) and housed in groups of 4–5 per cage. All mice were acclimatized to their environment one week before conducting experiments and provided free access to conventional standard chow and water. Following one week, mice were assigned randomly to specific experimental groups, were weighed before the commencement of experiments, and were provided with respective experimental semi-pure diets per group ad libitum (Semi-Pure Maintenance AIN93M diet (AIN93M), 23% Fat, High Glycaemic Index Modification of AIN93M (HFD), High Fibre Fruit and Vegetable powder Modification of AIN93M (SUP)). All mice were allowed to acclimate to their respective semi-pure diets for two weeks. Following acclimation to the semi-pure diets, the mice were taken through the experimental protocol (See Section 4.2 Experimental Design) and continually provided with the respective diets ad libitum. On completion of each experiment, all mice were anaesthetized and humanely killed. Cardiac blood, brain tissue, colonic segments and adrenal glands were collected and kept at −80 °C for further analyses.

All animal procedures followed the protocols approved by the Animal Ethics Committee of the University of South Australia.

### 4.2. Experimental Design

Sixty-three male 10–12-week-old C57BL6 mice were separated into experimental groups according to the respective semi-pure diets and either housed as controls with minimal human interaction or taken through a chronic unpredictable mild stress (CUMS) protocol (Control AIN93M = 10, control HFD = 10, control SUP = 10; CUMS AIN93M = 10, CUMS HFD = 12, CUMS SUP = 11). For the CUMS groups, chronic stress was induced by chronic exposure to different types of stressors daily such as cold swim (13 ± 1 °C, 5 min), warm swim (37 ± 2 °C, 5 min), overnight illumination (12 h), moist bedding (8 h), cage tilt 45° (8 h), food and water deprivation (24 h), tail pinch (60 sec, 1 cm from the end of the tail), reverse day and night (24 h), cage shaking (5 min) or no stress for four weeks. All stressors were applied randomly to prevent the habituation of mice to their repeated presentation. Briefly, each stressor was assigned a number sequentially and randomised using the RANDBETWEEN function in Microsoft Excel (Microsoft^®^ Excel^®^ for Microsoft 365 MSO (Version 2308 Build 16.0.16731.20182)). On completion of the CUMS protocol, behavioural testing was carried out the following day on all mice.

### 4.3. Behavioural Tests

All behavioural tests commenced at 9:00 a.m. on the day of each test. All mice were acclimatised to the testing environment for 10 min before each behavioural test.

#### 4.3.1. Open Field Test

Each mouse was placed in the same corner of a white plexiglass square apparatus (40 cm length × 40 cm width × 40 cm height) and allowed to freely explore the apparatus for a 5-min duration. Meanwhile, the activity of each mouse was recorded using an overhead camera and ANY-maze video tracking software (ANY-maze version 7.01, Stoelting, Wood Dale, IL, USA).

#### 4.3.2. Elevated Plus-Maze Test

The elevated plus maze comprises an elevated (40–70 cm from the floor) plus-shaped apparatus with two opposing closed arms with walls measuring 10 cm in height and two opposing open arms with no walls. Each mouse was placed in the centre of the maze and allowed to freely explore the maze for five minutes, during which each animal’s activity was recorded using ANY-maze video tracking software.

#### 4.3.3. Tail Suspension Test

Each mouse was suspended by its tail using sticky tape, and a camera was placed at a distance but perpendicular to the body in full view of the mouse. Each mouse’s response was recorded for 6 min. Immobility was considered as the mouse remaining completely motionless without struggling. The total immobility time was counted manually using a stopwatch and recorded.

### 4.4. Blood Sample Collection and Analyses

Non-fasting cardiac blood samples were obtained during tissue collection. Serum samples were allowed to separate in microtubes for 10 min before centrifugation at room temperature at 3000× *g* RPM for 8 min. Serum samples were stored immediately at −80 °C following centrifugation. Serum triglycerides and cholesterol were measured using the automated Indiko Plus Clinical Chemistry Analyser (Catalog no. 98640000, Thermo Fisher Scientific, Nedlands, Australia).

### 4.5. Fresh Tissue Collection and Homogenisation

Using a Precellys 24 Homogeniser (Bertin Technologies, Montigny-le-Bretonneux, France), tissues were homogenized in RIPA buffer (50 mM tris, 150 mM sodium chloride, 1 mM Ethylenediaminetetraacetic acid, 0.5% Triton X-100, 0.5% Sodium deoxycholate, pH 7.4) plus cocktail protease inhibitor (Sigma-Aldrich, St. Louis, MO, USA), thereafter homogenates were centrifuged at 13,000× *g* RPM for 30 min. The supernatants of the homogenates were collected, and the protein concentration was estimated using a bicinchoninic acid protein assay kit (Thermo-scientific, Rockford, IL, USA) according to the manufacturer’s instructions. The remaining supernatant was used for subsequent western blot analyses.

### 4.6. Western Blot

30 μg of protein for each hippocampal, prefrontal cortex, colon and adrenal sample was separated by gel electrophoresis on 12% SDS-polyacrylamide gels using the CBS gel system (C.B.S Scientific, San Diego, CA, USA) for 90 min at 120 V. Separated proteins were transferred overnight (960 min) at 100mA onto 0.2 or 0.45 μm nitrocellulose membranes (GE Healthcare Australia Pty Ltd., Sydney, New South Wales, Australia) and air-dried for 45 min. The air-dried membranes were blocked in 5% skim milk/TBST + 0.05% azide (Sigma-Aldrich) for 1 h at room temperature with gentle shaking followed by washing in TBST. The membranes were incubated overnight at 4 °C with primary antibodies (See Appendix B for details). Following overnight incubation with primary antibodies, membranes were washed in TBST and incubated for 1 h at room temperature with corresponding secondary antibodies for near-infrared Western blot detection (Li-Cor Biosciences, Lincoln, NE, USA). The resultant immunoblots were imaged using the Odyssey CLx imaging system (LI-COR Biosciences) and quantified using Image Studio Lite 5.2 (LI-COR Biosciences). Proteins were normalized with anti-β-actin or glyceraldehyde-3-phosphate dehydrogenase (GAPDH).

### 4.7. Histological Analyses

Colonic segments were collected using Swiss-rolling techniques published elsewhere [88], fixed in 10% neutral buffered formalin overnight at room temperature and embedded in paraffin. Histological damage was evaluated using 4-μm thick sections stained with haematoxylin and eosin. Colonic crypt length was evaluated using NPD.view v2.6.17 (Hamamatsu Photonics, Hamamatsu City, Japan) by measuring five colonic crypts demonstrating complete morphology over three fields of view at 20× magnification. It was expressed as the average of the total measurements. Goblet cells were visualised by staining 4-μm thick sections with a haematoxylin and period acid-Schiff (PAS) stain. Briefly, sections were deparaffinised and hydrated to deionised water, immersed in 1 g/dL periodic acid solution (Catalog no. 3951, Sigma) for 5 min at room temperature (18–26 °C), followed by immersion in Schiff’s reagent (Catalog no. 3952, Sigma, St. Louis, MO, USA) for 15 min at room temperature and counterstained in Ehrlich’s haematoxylin for 90 s. PAS-stained goblet cells were counted across ten colonic crypts.

### 4.8. Faecal Sample Collection and 16S rRNA Sequencing

Samples were stored at −80 °C, and microbial DNA extraction, amplification, sequencing, and diversity profiling were performed by the Australian Genomics Research Facility (AGRF, Melbourne, Australia). Briefly, the 16S rRNA genes (V3-V4) were amplified using 341F (CCTAYGGGRBGCASCAG) and 806R (GGACTACNNGGGTATCTAAT) primers. Next Generation Illumina sequencing was employed for amplicon sequencing of the DNA library by AGRF, and 300-bp paired-end reads were generated. Primary image analysis was performed using MiSeq Control Software v3.1.0.12 and Real-Time Analysis v1.18.54.4. Diversity profiling was performed with Quantitative Insights into Microbial Ecology (QIIME 2 2019.7) [89]. The demultiplexed raw reads were primer trimmed and quality filtered using the cutadapt plugin, followed by denoising with DADA2 by AGRF [90]. Taxonomy was assigned to amplicon sequence variants using the q2-feature-classifier classify-sklearn naïve Bayes taxonomy classifier. The SILVA rRNA database [91,92] was used to annotate taxonomic information for the representative sequences of the OTUs.

### 4.9. Microbiota Analyses

The sequenced 16S rRNA data provided by AGRF was further analysed using QIIME 2 (2022.2). These data were rarefied to a depth of 23,700 sequences before analysis of α- and β-diversity to ensure even sampling with 100% retention of samples. α-Diversity was assessed by the number of observed features in observed OTUs to determine species richness within each sample and by the Simpson index to determine species evenness. The Bray-Curtis β-diversity index was used to determine dissimilarities between microbial populations between samples. Differences in relative abundance were evaluated at the phylum level to identify major shifts in the gut microbiome. Following ANCOM analysis using QIIME 2 (2022.2), differences in relative abundance were evaluated at the genus level.

### 4.10. Statistical Analyses

Two-way ANOVA analyses were conducted using PRISM v8.3 (GraphPad Software, Inc., San Diego, CA, USA) with treatment as a between-subject factor. The Shapiro-Wilk normality test was used to determine the data distribution before analysis. Post hoc multiple comparisons were made with Bonferroni adjustments. Data transformations were carried out before two-way ANOVA for the serum triglyceride data and the data for the time spent in the central zone of the OFT since these deviated from the Gaussian distribution. All other data followed the Gaussian distribution. The α-diversity Observed OTUs and Simpson Index were compared between the control and treatment group, taking each diet into account, using two-way ANOVA. β-diversity index comparisons were made using PERMANOVA tests. Statistical significance was set at *p* < 0.05.

## 5. Conclusions

The present study showed that four weeks of CUMS induced anxiety-like behaviour in mice on a semi-pure AIN93M diet but could not robustly induce depressive-like behaviour. Mice on this diet exposed to the CUMS protocol showed increased SERT and IDO protein levels, suggesting increased re-uptake of serotonin and up-regulated serotonin metabolism via the kynurenine pathway in both the hippocampus and PFC. In the colon, these mice also displayed a decrease in colonic goblet cells, potentially reducing the protective mucus barrier of the gut, possibly contributing to colonic inflammation and up-regulating tryptophan metabolism along the kynurenine pathway. Despite these changes in the colon, CUMS could not reduce the diversity of the gut microbiota, although future studies are needed to confirm this effect. This study also provided new data on the adrenal medulla response to chronic stress since CUMS reduced the catecholamine synthetic capacity of TH in the adrenal glands, which may be responsible for the blunted sympathoadrenal response to acute stress seen in anxiety disorders and depression. Key findings of the impacts of high-fat, high-GI and high fibre, fruit & vegetable “superfood” powder modifications of the AIN93M diet on the effects of CUMS are summarized in Box 1.

Box 1A summary of the effects of dietary modifications of the AIN93M diet on behaviour in mice on a 4-week chronic mild unpredictable stress protocol.
**A high-fat, high-GI semi-pure diet:**
It may act as a “comfort food” since it prevented anxiety- and depressive-like behaviours following CUMSReduced colonic SERT levels suggest reduced serotonin reuptake following CUMS, potentially exacerbating colonic inflammation.Decreased colonic goblet cell numbers following CUMS, similar to the AIN93M diet

**A high fibre, fruit & vegetable “superfood” semi-pure diet:**
Lowered cholesterol levels below the reference range for C57Bl6 miceIncreased anxiety-like behaviour is likely due to the inclusion of psyllium husk in the dietIt had a protective effect on the mucus barrier of the colon by preventing a reduction in goblet cell number following CUMSThese protective effects on the colon may have been driven by an increase in the species evenness of the gut microbiota and/or an increase in the abundance of *Bifidobacteria*Reduced the catecholamine synthetic capacity of tyrosine hydroxylase in the adrenal glands, possibly via upregulation of adrenal serotonin reuptake; however, the mechanisms governing this interaction require further study


## Data Availability

The data that support the findings of this study are available from the corresponding author upon reasonable request.

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
