# Peer review of "The Effects of Chronic Unpredictable Mild Stress and Semi-Pure Diets on the Brain, Gut and Adrenal Medulla in C57BL6 Mice"

_ijms, 2023, doi:10.3390/ijms241914618_

Round 1
Reviewer 1 Report
September 6, 2023
Manuscript IJMS-2606020
The Effects of Chronic Unpredictable Mild Stress and Semi-Pure 2 Diets on the Brain, Gut and Adrenal Medulla in C57BL6 Mice.
I suggest that all figures omit the **** symbolism (below the graph), which does not represent anything or is not understood.
On page 5, lines 144-147, they mention that the SUP diet decreases triglyceride levels, and the graph shows the opposite.
The symbolism of the charts could be more explicit.
Why do some graphs display data above the columns and others do not? Also, what are they? Significances?
I would remove the titles of graphs 1 and 2 and put panels since the y-axis shows what was evaluated and its units.
In Graph 1, is panel b necessary?
On page 11, lines 277-280, they mention that there is significance, but they do not put it in the graph. TPH2 in CUMS control vs. What? And interaction with what?
Also, isn't it repetitive with what it says on lines 279-280?
The last sentence of that paragraph states that SUP increases TPH2 despite the treatment, and I don't see that in the graph.
With more SERT, there would be less serotonin in the synaptic space, right?
Would that improve the conditions caused by CUMS?
How could it be the memory? If so, how do you explain the SERT graph in Figures 6 and 7?
How can the HFD diet increase the synthesis of serotonin?
Figure 7, TPH2 graph. Could the PFC mBDNF levels due to the SUP diet be considered compensatory?
The description of the images in Figure 10 could be improved. For example, in a) and b) and in section b), put the names of the treatments and do not follow the nomenclature sequentially.
How is the serotonin synthesized and used by the microbiota evaluated with brain function and see its effects on behavior?
The information is too much and complex. It could be made more reader-friendly.
The comments are in the corrected file v2

Reviewer 2 Report
This observational study investigated the effects of HFD and SUP diets on brain, gut, and adrenal medulla in CUMS mice. The results showed that CUMS induced anxiety-like behaviour, dysregulated tryptophan and serotonin metabolic pathways in the hippocampus, prefrontal cortex, and colon, altered the composition of the gut microbiota, and reduced synthetic capacity of catecholamine in the adrenal glands. HFD and SUP diets exerted differential effects on these parameters. Below are my comments:
1. Introduction: There was no study hypothesis. Please elaborate study hypothesis in the last paragraph of the Introduction.
2. Results: There were inconsistent results on the blots. Please indicate single values in all figures. Please also be specific in whether the data distributions in each figure were normal or not, and the statistical methods applied for the data not normally distributed.
3. Methods: There were no information on the randomized procedures adopted for the CUMS model. It is unclear whether the stressors were applied randomly or not.
4. Methods: There were no information on how to quantify the amount of diets taken by the mice. It is unclear whether the differential effects were derived from the components or amounts of diets.
5. Discussion: Please use one paragraph to discuss the clinical implications of the present study findings.
6. For the transparency, please indicate the number of study protocol approved by the Animal Ethics Committee, as well as the funding supported by the University of South Australia.
Minor editing of English language required.
Round 2
Reviewer 1 Report
I congratulate you on the effort to improve the manuscript substantially. Now, all the results and the interpretation of their findings are much more precise.
Author Response
The authors would like to thank Reviewer 1 for their review of the manuscript.
Reviewer 2 Report
1. Please indicate individual result values directly in figures.
2. Please be specific in the randomized procedures adopted for the CUMS model.
3. Was this study able to quantify diet amounts taken by each mouse? If not, it should be considered as a limitation.
